# Hyperintense Brain Lesions in Asymptomatic Low Risk Patients with Paroxysmal Atrial Fibrillation Undergoing Radiofrequency Pulmonary Vein Isolation

**DOI:** 10.3390/jcm10040565

**Published:** 2021-02-03

**Authors:** Joanna Wieczorek, Katarzyna Mizia-Stec, Anetta Lasek-Bal, Piotr Wieczorek, Iwona Woźniak-Skowerska, Anna M. Wnuk-Wojnar, Krzysztof Szydło

**Affiliations:** 1First Department of Cardiology, School of Medicine in Katowice, Medical University of Silesia, 40-635 Katowice, Poland; kmiziastec@gmail.com (K.M.-S.); piotr.i.wieczorek@gmail.com (P.W.); iskowerska@hoga.pl (I.W.-S.); awojnar@sum.edu.pl (A.M.W.-W.); kszydlo1964@gmail.com (K.S.); 2Department of Neurology, School of Medicine in Katowice, Medical University of Silesia, 40-635 Katowice, Poland; balanett@poczta.onet.pl

**Keywords:** atrial fibrillation, cognitive decline, magnetic resonance imaging, mini-mental state examination, white matter hyperintensities

## Abstract

Background: The aim was to determine the occurrence, consequences and risk factors for brain white matter hyperintensities (WMH) assessed in magnetic resonance imaging (MRI) in low-risk patients with paroxysmal atrial fibrillation (AF) undergoing radiofrequency pulmonary vein isolation (PVI-RF). Methods: 74 patients with AF (median 58.5 years (IQR 50–63), 45 male) were included. Before and after a minimum of 6 months after PVI-RF, a brain MRI and a mini-mental state examination (MMSE) were performed. Results: Baseline WMH lesions were found in 55 (74.3%) patients and in 48 from 62 (77.4%) patients after PVI-RF. The WMH lesions were more frequent among older patients, with a higher CHA2DS2-Vasc (C—Congestive heart failure/LV dysfunction, H—Hypertension, A—Age, D—Diabetes mellitus, S—Stroke, V—Vascular Disease, Sc—Sex category). Factors affecting the severity of the WMH were: older age, the co-existence of the PFO and coronary artery disease (CAD). After a follow-up period, the factors predisposing to brain WMH lesions occurrence (age, higher BMI and CHA2DS2-Vasc score) and to the more advanced changes (age, higher CHA2DS2-Vasc score, CAD, PFO) were obtained. Conclusions: The presence and severity of cerebral microembolism are associated with age, higher CHA2DS2-Vasc score and the coexistence of PFO and CAD. PVI-RF procedure and its efficacy does not influence on MRI lesions. In this population, cerebral microembolism is not related to cognitive impairment.

## 1. Introduction

Atrial fibrillation (AF) leads to the formation of microemboli embolizing cerebral microcirculation [1]. Because patients may be asymptomatic for a long time, these changes are considered clinically silent. Chronic cerebral microembolism associated with AF and hypoperfusion of cerebral circulation gradually lead to the development of clinically silent cerebral ischemia (SCI), which may be the basis for the development of neuropsychological deficits and even lead to dementia.

SCI refers to small vessel disease and, next to TIA and stroke, is included in the entire panel of cerebral vascular diseases [2]. Importantly, the clinical significance of SCI is highlighted by the fact that their incidence is up to 10 times higher than that of strokes [3]. Until now, brain lesions have been consistently associated with small vessel disease, a higher incidence of dementia, and impaired global cognitive function [4,5,6]. Therefore, small vessel disease based on structural changes such as silent brain infarcts, white matter hyperintensities (WMH), and brain micro-bleeding can be a key link between AF and cognitive disorders, especially in neurologically asymptomatic patients. The importance of AF in the context of brain changes and cognitive impairment should be considered, taking into account mainly SCI. Ischemia occurs in the mechanism of microembolism of small cerebral vessels with thrombi formed during turbulent blood flow in the left atrium (LA).

The division of WMH type brain lesions, found in the magnetic resonance imaging (MRI) study, proposed by Fazekas et al. in 1987, and reproduced by many subsequent researchers [7,8,9] has become common. In clinical conditions, visual scales are preferentially used to assess WMH of the brain, taking into account shape, severity or location of hyperintensive brain lesions, while the scale proposed by Fazekas et al. is considered as a reference. The modified Fazekas scale is adequate to describe the intensity and range of WMH foci described in the MRI study [10].

These changes are often observed in patients with AF, but the relationship between them and AF is not clear, especially in young people. The mere presence of WMH lesions in the MRI of the brain is potentially related to AF and hypertension. Comorbid risk factors for vascular diseases such as older age, obesity, hyperlipidemia, diabetes, lower physical activity and generalized inflammatory response can also play an important role [11,12,13,14,15,16]. These changes may be associated with observed neuropsychological, motor and mood disorders. The potential role of AF in cognitive impairment and dementia due to cerebral microembolism (imaged as WMH changes in the brain MRI) has become of recent interest. Assessment of the patient’s overall neurocognitive function can be performed using the mini-mental state examination (MMSE) test, the most common and simple tool designed to test a wide range of cognitive functions. The scale can also be helpful in fast screening for dementia exclusion.

Embolic complications leading to clinically significant stroke represent a small percentage of all complications associated with percutaneous pulmonary vein isolation (PVI). According to available data from large world registers, the risk of periprocedural stroke is 0.5%–1% [17,18,19,20,21]. However, considering the significance of PVI in the context of clinically silent microembolic changes, it has been proven that depending on the source of energy used, the risk of their occurrence is high. The frequency of reported ischemic lesions in MRI of the head the day after ablation ranges from 4.3% to 50% [22,23,24,25,26,27]. In the case of percutaneous PVI procedures using radiofrequency pulmonary vein isolation (PVI-RF) of the open-irrigated type, microembolic complications after the procedure were observed in 5%–18% of cases [23,24,25,26,27,28,29]. In turn, after the PVI-RF duty-cycled phased procedure, the reported incidence of cerebral ischemic foci in MRI studies is significantly higher—from 37.5% to 38.9% [22,23].

Most brain imaging examinations for peri-procedural silent ischemic lesions were performed 24–48 h after ablation. However, there is still a lack of reliable data on the long-term effects of the PVI-RF procedure in relation to the presence of cerebral ischemic foci and their potential impact on cognitive function in relation to the effectiveness of the procedure.

### The Aims of the Study Were


To determine the occurrence, consequences and risk factors for brain white matter hyperintensities (WMH) assessed in magnetic resonance imaging (MRI) in low-risk patients undergoing PVI-RF.To determine risk factors for brain WMH lesions assessed in the brain MRI in low-risk patients before and after PVI-RF.To determine the impact of PVI-RF procedure on the occurrence and severity of WMH lesions assessed in the brain MRI.To assess a potential relationship of atrial fibrillation with cognitive decline, with particular relation to PVI-RF impact.


## 2. Materials and Methods

Seventy-four patients with diagnosed paroxysmal non-valvular AF, who were hospitalized in the Department of Cardiology between 2013 and 2017 in order to perform PVI for the first time, were enrolled into the study. The study group was evaluated during hospitalization before the procedure and after a minimum of 6 months after the PVI-RF procedure the patients were invited for clinical re-evaluation. The study group consisted of patients with a median age of 58.5 (IQR (interquartile range) 50–63 years) with a predominance of male (60.8%). Patients with AF diagnosed up to 5 years, symptomatic (median EHRA scale 3) and relatively low score on the CHA2DS2-Vasc scale (median scale 2) predominated.

The standard inclusion criteria were: documented paroxysmal symptomatic non-valvular atrial fibrillation (EHRA IIb-IV), despite the optimal treatment, and qualified for the PVI, adequate warfarin treatment before admission, maintaining sinus rhythm during hospitalization before enrollment, preserved LV systolic function (LV EF ≥ 50%), written informed consent and >18 years of age.

We excluded patients with: a history of artery pathology (uni- or bilateral carotid artery, ascending aorta, carotid or vertebral artery atherosclerotic stenosis (defined as arterial stenosis ≥ 50% in the NASCET (North American Symptomatic Carotid Endarterectomy Trial) [11] or dissection; Behçet disease; uni- or bilateral intracranial artery stenosis; a history of carotid angioplasty or endarterectomy; vasculitis), connective tissue disease, neuroborreliosis, multiple sclerosis, Sneddon syndrome, a history of stroke or transient ischemic attack (TIA), structural heart disease (cardiomyopathies, significant valvular heart disease), states connected with hypercoagulation or a predisposition to systemic embolism, a history of PVI, pregnancy, refusal to participate, acute kidney disease or kidney disease chronic with glomerular filtration (GFR) < 30 mL/min, contraindications for MRI.

Informed written consent was obtained from each patient. The study protocol was approved by the Bioethical Committee of the Medical University of Silesia in Katowice in Poland (ethical approval number KNW/0022/KB1/72/13 26.06.2013) and performed according to the ethical guidelines of the 1975 Declaration of Helsinki.

On admission, a detailed medical history that included the current course of the disease, the main symptoms (classified in EHRA class), concomitant diseases (including coronary artery disease, type 2 diabetes, arterial hypertension, hyperlipidemia, peripheral artery disease), a familial history of arrhythmia, current pharmacotherapy (especially compliance using oral anticoagulants) and tobacco smoking was collected from each subject. We also collected physical examination parameters: weight, height and body mass index (BMI), body surface area (BSA). During hospitalization laboratory tests, an MRI of the head, MMSE test, 24-h ECG Holter monitoring, transthoracic and transesophageal echocardiography, as well as Doppler ultrasound of extracardiac carotid and vertebral arteries were performed in all patients.

A minimum 6-month follow-up period of medical history, including possible recurrence of arrhythmia and pharmacotherapy, was noted. Furthermore, in all subjects, we performed: transthoracic echocardiography, MRI of the head, MMSE test and 7-day ECG recording using the Holter method.

### 2.1. Magnetic Resonance Imaging

All participating patients underwent an MRI of the brain. The investigation was performed the day before the ablation using a 1.5-T scanner (Siemens Healthcare GmbH, Erlangen, Germany). MRI was performed using the standard sequences: T1, T2, FLAIR, SWI, DWI and 3DFLAIR. No contrast dye was used. All MRI images were analyzed independently by an experienced radiologist and neurologist, both of whom were blinded to the clinical status of the patients. Brain WMH size and quantity were assessed using the Fazekas scale and divided into three degrees of severity:Grade 1—mild WMH were defined by punctate lesions with a maximum diameter of 9 mm for a single lesion and of 20 mm for grouped lesions.Grade 2—moderate WMH were early confluent lesions of 10–20 mm single lesions and >20 mm grouped lesions of any diameter and only connecting bridges between the individual lesions.Grade 3—severe WMH were single lesions or confluent areas of hyperintensity ≥ 20 mm in diameter.

### 2.2. Test Mini-Mental State Examination

The MMSE test was performed in a secluded place using the version according to Folstein et al. [30], recommended by the Interdisciplinary Group of Experts in the Diagnosis of Dementia of the Psychogeriatrics and Alzheimer’s Disease Section of the Polish Psychiatric Association. The MMSE test evaluated: orientation in time and place, recall, attention and calculation, language manipulation and constructional praxis. The maximal test result is a score of 30, and a 27–30 result is correct. A score less than 24 is highly suggestive of the presence of cognitive decline or dementia. The result of 24–26 suggests mild cognitive impairment.

### 2.3. ECG Holter Monitoring

Holter ECG recordings were made using Lifecard CF recorders (Spacelabs Healthcare, Snoqualmie, United States) and were analyzed using the Sentinel software (version 11, Del Mar Reynolds, Irvine, United States). The registration was made during the day immediately preceding the PVI-RF procedure and during the control examination at least 6 months after the procedure. Registration after the observation period was carried out using the 7-day option with ECG recording at home. The analysis of ECG records included an assessment of the rhythm frequency (minimum, maximum and average), as well as supraventricular and ventricular arrhythmias. An episode of AF was considered to be arrhythmia lasting > 30 s, two episodes of AF at the same time separated by sinus rhythm lasting < 30 s were considered as one episode.

### 2.4. Transthoracic/Transesophageal Echocardiography and Carotid Ultrasound

On admission, ECG-gated transthoracic and transesophageal echocardiography, as well as a carotid artery Doppler ultrasound was performed in all patients. An experienced physician took all of the measurements using the same investigation protocol and techniques in order to reduce inter- and intra-observer variability. The echocardiography investigation was performed using a GE Healthcare VIVID 7 Dimension (General Electrics Medical Systems, Horten, Norway) with a 2.5 MHz sector ultrasound transducer for transthoracic, while a 2–7 MHz for transesophageal echocardiography. The study transesophageal echocardiography (TEE) assessed the patency of PFO and the presence of thrombi and echogenic blood in the LA appendage (grade 1 to 3). Using pulsed doppler, blood flow velocity in the LA appendage was measured. Patients with the presence of thrombi and third grade echogenic blood in the LA appendage were excluded.

An extracranial artery Doppler ultrasound was performed using a GE Healthcare VIVID 7 Dimension (General Electrics Medical Systems, Horten, Norway) with the visualization of the common, internal, external and vertebral carotid arteries was performed by experienced ultrasonographist using a 10 MHz linear transducer in order to exclude significant extracranial artery stenosis.

### 2.5. PVI-RF Procedure

Access to pulmonary veins was obtained using the Seldinger technique. At the beginning of the procedure, rotational LA angiography (injection of contrast agent into the pulmonary artery during rapid right ventricular stimulation) was performed. Then, after transseptal puncture, 3-dimensional electro-anatomical mapping was performed using the CARTO^®^3 system (Biosense Webster, Diamond Bar, CA, USA). Pulmonary vein isolation was obtained by RF radiofrequency ablation with a ThermoCool^®^ SmartTouch^®^ catheter (Biosense Webster, Diamond Bar, CA, USA). The procedure was performed using a Lasso electrode (Biosense Webster, Diamond Bar, CA, USA) or Achieve (Medtronic, MN, USA).

The procedure was performed at the international normalized ratio (INR) assessed on the day of ablation from subtherapeutic values to 2.5. Immediately after transseptal puncture, all patients received an intravenous bolus of unfractionated heparin (100 IU/kg), followed by a continuous infusion of unfractionated heparin (2000 IU/h) through a sheath inserted through the atrial septum to obtain an activated clotting time (ACT) > 300 s. During the procedure, the ACT was measured at 30-min intervals. In patients with INR on the day of ablation less than 2, after the procedure, a 24-h intravenous continuous infusion of unfractionated heparin was performed under the control of APTT (activated partial thromboplastin time). The next dose of vitamin K antagonists (VKA) was given 4 h after ablation to get an INR of 2.0 to 3.0 the next day.

### 2.6. Statistical Analysis

Statistical analysis was performed using STATISTICA software (version 13.1 PL, TIBCO Software Inc., Palo Alto, California, United States). All data were collected in a Microsoft Office Excel spreadsheet (version 2016 PL, Microsoft, Redmond, United States). A *p* value of less than 0.05 was considered to be statistically significant. Results for continuous variables are presented as the mean with standard deviation for normal distributions or the median with interquartile range for non-normal distributions. The normality of the distribution of continuous variables was verified with the Shapiro–Wilk test. The tables also present ordinal variables as percentages of the trait frequency. Depending on the distribution of variables, parametric and non-parametric tests were used for dependent and independent variables.

## 3. Results

### 3.1. The Study Group Characteristic

Table 1 presents the basic parameters characterizing the study group, the results of the TEE examination and comorbidities and factors considered to be cardiovascular risk factors. PFO was found in 20 patients (27%). The study group was dominated by patients with no echogenic or current grade 1 blood. Patients with LA appendage thrombus were excluded from the study. In the study group, just over half of the patients had hyperlipidemia and hypertension.

Baseline pharmacotherapy in the study group included a wide group of cardiovascular drugs. Acetylsalicylic acid was taken by 1 patient (1.4%), angiotensin converting enzyme inhibitor—34 (45.9%), AT1 receptor antagonist—10 (13.5%), beta-blocker (non-sotalol)—52 (70.3%), calcium channel blocker—9 (12,2%), statin—35 (47.3%), fibrate—4 (5.4%), diuretic—19 (25.7%), sotalol—12 (16.2%), amiodarone—9 (12.2%), propafenone—27 (36.5%). All patients were taking vitamin K antagonist (VKA; acenocumarol—52 (70.3%), warfarin 22 (29.7%)). No patients were taking long-acting nitrates or clopidogrel.

### 3.2. WMH Lesions and Psychological Assessment before PVI-RF Treatment

There were no significant differences in the results of MMSE test depending on brain WMH lesions presence and severity. Patients without pre-PVI-RF brain WMH lesions obtained a median of 30 points in the MMSE test (IQR 28–30), while 29 points (IQR 28–30) when lesions were present (*p* = 0.11). Table 2 presents MMSE test scoring depending on the pre-PVI-RF procedure WMH lesions severity.

Depending on the pre-PVI-RF presence of any WMH lesions in the brain MRI (lesions absent or present regardless of severity), patients differed in age and obtained CHA2DS2-Vasc score. Patients with present WMH lesions were older (60 (56–65) vs. 44 (38–56), *p* < 0.001) and scored higher on the CHA2DS2-Vasc score (2 (1–3) vs. 1 (0–2), *p* = 0.009), when compared with patients without any WMH-type lesions.

Multivariate logistic regression analysis was performed with the inclusion of the following pre-PVI-RF procedure variables: CHA2DS2-Vasc, AF duration time, BMI, PFO presence, echogenic blood presence in the left atrium appendage, tobacco smoking and hyperlipidemia. CHA2DS-Vasc was statistically significant (*p* = 0.02). In the second one, the multivariate logistic regression analysis model, the variables included in the CHA2DS2-Vasc scale were analyzed: arterial hypertension, age, diabetes mellitus, sex, coronary artery disease and peripheral artery disease. Patients with congestive heart failure were excluded from the study. Age was statistically significant (*p* < 0.001).

Depending on the pre-procedural evaluation of WMH lesions severity in the brain MRI (lesions absent or present in the Fazekas scale: 1–3), statistically significant patients differed in age, the CHA2DS2-Vasc scale score, presence of PFO and coronary artery disease (Table 2).

### 3.3. WMH Lesions and Psychological Assessment after PVI-RF Treatment

There were no significant differences in the results of MMSE test depending on brain WMH lesions presence and severity. Patients without follow-up brain WMH lesions obtained the median of 30 points in the MMSE test (IQR 29–30), while 29 points (IQR 28–30) when lesions were present (*p* = 0.1). Table 3 presents MMSE test scoring, depending on the WMH lesions severity after the observation period.

Patients with any brain WMH lesions assessed after the post-PVI observation period were older (61 (56–65) vs. 45 (39–56), *p* < 0.001), had a higher BMI (29.4 ± 4.7 vs. 27.4 ± 2), *p* = 0.009), had a higher score on the CHA2DS2-Vasc scale (2 (1–3) vs. 1 (1–2), *p* = 0.02), as well as a lower pre-procedural MMSE score (28 (27–29) vs. 29.5 (29–30), *p* = 0.002) compared with patients without WMH lesions.

Multivariate logistic regression analysis was performed with the inclusion of the following post-PVI-RF procedure variables: CHA2DS2-Vasc, AF duration time, BMI, PFO presence, echogenic blood presence in the left atrium appendage, tobacco smoking and hyperlipidemia. CHA2DS-Vasc was statistically significant (*p* = 0.02). In the second one, the multivariate logistic regression analysis model, the variables included in the CHA2DS2-Vasc scale have been analyzed: arterial hypertension, age, diabetes mellitus, sex, coronary artery disease and peripheral artery disease. Patients with congestive heart failure were excluded from the study. Age was statistically significant (*p* < 0.001).

Depending on the WMH lesions severity in the brain MRI, patients statistically significantly differed in terms of age, CHA2DS2-Vasc scale score and coronary artery disease presence. A trend towards a difference in the incidence of PFO was obtained (Table 3).

### 3.4. WMH Lesions and Psychological Assessment Depending on PVI-RF Procedure

Median follow-up was 9.9 months (IQR 7.6–11.8 months). In the 7-day Holter evaluation after the observation period, the effectiveness of the PVI-RF procedure was confirmed in 53.8%.

In the study group, no statistically significant differences were found between the presence and severity of the brain WMH lesions, as well as MMSE scores before and after the PVI-EF observation period (Table 4). There were no statistically significant differences between the presence and severity of post-PVI-RF WMH lesions in the brain, as well as the post-PVI-RF MMSE score in patient subgroups depending on PVI-RF procedure effectiveness.

## 4. Discussion

This publication represents part of a single-center, non-randomized, prospective study of a population consisting of relatively young patients, with a history of paroxysmal, symptomatic AF (median EHRA 3), without significant structural heart disease and with a low score obtained in the CHA2DS2-Vasc scale that were classified to PVI-RF procedure.

### 4.1. Study Group

The inclusion of only older patients could not only affect the structural changes of the heart related to its fibrosis or the accumulation of cardiovascular risk factors, but also prevent a reliable assessment of the presence and severity of WHM lesions in the brain. In order to avoid the accumulation of the brain WMH risk factors, patients with a relatively low CHA2DS2-Vasc score dominated in the whole group. Stable coronary artery disease was diagnosed in 15 patients (19% of the study group), while the study turned out that risk factors for atherosclerosis and coronary artery disease were not related to the effectiveness of PVI-RF procedure. In the study, the main criterion for inclusion in the study was LV systolic function, i.e., LV EF ≥ 50%.

To ensure the homogeneity of the group, it was limited to patients with paroxysmal AF, especially because this group benefits most from the PVI-RF procedure in the context of effectiveness [31,32,33]. Therefore, the study group consisted of patients with paroxysmal AF, for the most part with an established arrhythmia diagnosis < 5 years (46% of patients), and the duration of AF had no effect on the post-observation effectiveness of the PVI-RF procedure. Finally, the subgroups of patients with successful and ineffective PVI-RF procedure after observation period were homogeneous in terms of sex, age, and the burden of associated diseases.

Due to the longer healing and remodeling process of LA, as well as the longer time needed to assess WMH lesions and cognitive impairment, the minimum period after which patients were re-evaluated was 6 months. In the available literature, most studies showing silent peri-procedural brain microembolic changes were performed 24–48 h after surgery for 3 months, rarely after a longer period of time. There is still a lack of reliable data on the long-term effects of the PVI-RF procedure in relation to the presence of WMH lesions of the brain and their potential impact on cognitive function, with particular emphasis on the effectiveness of the procedure.

### 4.2. WMH Lesions before and after the PVI-RF Procedure

According to various data, even 60% of patients with AF may have asymptomatic brain changes on a microembolic basis, regardless of the procedures performed in the left heart chambers [1,34]. These changes in the white matter of the brain in patients with AF appear to be secondary to concomitant disease processes, and their relative size is much higher compared to the minimum number and size of the lesions caused by cardiovascular procedures in this group of patients. By assessing pre-PVI-RF MRI images for the presence of WMH lesions and their relationship to clinical parameters, statistically significant more frequent occurrence of WMH cerebral lesions was found among older patients and those achieving a higher CHA2DS2-Vasc score. While MRI images in patients with current WMH lesions in terms of their severity, patients with more advanced lesions were older, had higher CHA2DS2-Vasc score, PFO and coronary artery disease. After the minimal observation period, no statistically significant differences were found among the other assessed parameters.

In the available literature, most brain imaging studies for microembolic lesions after PVI procedures were performed using 1.5 Tesla MR devices. At present, the latest reports using 3-Tesla MR devices indicate the possibility of more frequent occurrence of silent microembolic changes in the brain in the group of patients with AF [35]. However, the reported frequency of microembolic lesions after ablation procedure due to AF may vary depending on the definition of diagnosis, methodology and used ablation technique. Therefore, it is difficult to clearly identify potentially modifiable factors common to all literature data, indicating the cause and affecting the frequency of brain WMH lesions.

The mechanism of brain WMH lesions in patients undergoing PVI-RF ablation remains poorly understood. It is certainly multifactorial and seems to be a consequence of the ablation procedure itself. While it seems clear that these lesions are the result of gas or particles brain tissue microembolization, consideration of potential confounding factors should also include aspects of the patient or the equipment used during the procedure. The presence of brain microembolic changes as a result of intracardiac procedures requiring access to the left chambers of the heart has become a topic indicating the thrombogenic potential of various technologies and ablation approaches. In available literature, irrigated RF ablation techniques dominate in descriptions of PVI techniques, using single-tip catheters. A similar technique was used in the present study.

During PVI ablation due to AF, heparinization with monitoring and maintenance of minimum ACT time values of 300 or 350 s is recommended. Appropriate heparinization may reduce the formation of thrombi on catheters and transseptal sheaths located in LA, as well as at the ablation site. Scaglione et al. showed that peri-operative ACT time below 320 ms is an independent prognostic factor of brain emboli changes during the observation period [36]. In another study conducted by Martinek’s team, there was no difference in minimal and medium levels of ACT when comparing patients with peri-procedural cerebral microembolic lesions [37]. In own work, during the PVI-RF procedure, the goal was to achieve ACT > 300 ms as standard.

In this study, patients were prepared for ablation with anticoagulant treatment with the VKA drugs group, in accordance with the ESC recommendations in force at that time [38]. In the light of recent reports, the need for uninterrupted periprocedural anticoagulant treatment, both VKA and factor X inhibitors, is emphasized, which may be important in reducing periprocedural cerebral microembolic events [26,37]. It has been shown that the continuation of VKA treatment with peri-procedural maintenance of the therapeutic INR ratio ≥ 2.0 is associated with a significantly lower incidence of post-ablation brain microembolic lesions. On the other hand, subtherapeutic INR levels were associated with a 3.1-fold higher risk of microembolism [26]. In the current study, the average INR determined on the day of ablation in the study group was 1.9 on average. There was no correlation between the periprocedural INR value and the presence and severity of WMH lesions in the brain, before and after PVI-RF. In the literature, can be found single reports on the potential impact of ACT time obtained during the ablation procedure [26,36,37], intra-procedural electrical cardioversion [26,37], the presence of self-contrasting blood in the LA appendage in the pre-treatment TEE [36], scoring CHA2DS2-Vasc scale [39], age [37] or pre-ablation brain WMH lesions [25] for an increased incidence of post-procedural microembolic lesions in the brain. In our own work, these reports were partially confirmed, although after several months of observation period a similar frequency and severity of WMH lesions in the brain MRI were found, compared to the results obtained before the PVI-RF procedure.

### 4.3. Psychological Assessment before and after the PVI-RF Procedure

Scientific evidence suggests that AF alone is associated with a higher risk of cognitive impairment and dementia, even in patients with no stroke history [39,40]. It is associated with more than doubling the risk of developing silent brain damage [41]. It is believed that silent cerebral ischemia in patients with AF arise on a microembolic basis. Due to their small size and location (away from speech and motor centers), they usually do not cause clinically apparent focal neurological deficits. However, with the accumulation of silent brain microembolic lesions, a progressive cognitive deficit may occur [42]. Our own results do not confirm these premises. While the incidence of silent brain damage in the form of WMH described in MRI of the brain is high in the study group (68.8%), patients with baseline WMH-type lesions (taking into account their severity) did not achieve worse results in the MMSE test as compared to patients with normal brain image on MRI. Perhaps the study group characteristic, including the relatively young age of patients, explains such observations.

Another issue is the impact of PVI-RF ablation in AF patients on cognition by generating new microemboli foci. Most of the indications that the PVI-RF procedure itself may affect cognitive function come from observational, non-randomized studies. So far, the relationship between silent cerebral embolism and cognitive impairment in patients undergoing PVI-RF has not been clearly confirmed. In this study, it was assumed that the selected population (taking into account numerous selection criteria and excluding patients after symptomatic stroke) is relatively “young” and not burdened with many comorbidities. In all patients, except brain MRI, we performed a parallel neuropsychological assessment, taking into account the effectiveness of the PVI-RF procedure. In the study group, there was no significant difference between the initial cognitive abilities assessed at baseline using the MMSE test, and the test results obtained after the observation period. Similarly, patients who underwent successful PVI-RF procedure obtained a comparable result in the MMSE test, in relation to patients with recurrent AF after the observation period.

Patients with reported WMH lesions after the follow-up MRI showed significantly lower results in the MMSE test performed before the PVI-RF procedure, which seems to be accidental. There were no significant differences in the MMSE test results depending on the presence and severity of WMH changes in brain image analysis after the observation period. For comparison, in the prospective MACPAF study, 37 patients with paroxysmal AF underwent MRI cerebral imaging (3-Tesla device) within 48 h after PVI [43]. New microembolic lesions were found in 43.2% of patients, of which only 6.5% of acute post-procedural brain lesions in MRI scans after 6 months constituted a permanent scar. Neuropsychological assessment after 6 months showed no significant effect of these changes on attention, motor function, short-term memory or learning.

Although in the case of PVI procedures, the periprocedural frequency of new microembolic lesions is usually estimated to be 18% [23,26,28,29]; the long-term consequences of asymptomatic or subclinical ischemic stroke associated with the procedure are unknown. Cognitive functions are not routinely tested after ablation procedures, and the mechanisms underlying these disorders are still not fully understood.

### 4.4. Limitations

The analyzed group was relatively small and the statistical power of this study is limited. The population consisted of a selected group of patients that were quite young and had a low thrombo-embolic risk and no significant concomitant diseases. The conclusions should be limited to populations of asymptomatic patients with non-valvular paroxysmal AF. On the other hand, the characteristics of the study group make the results interesting. Increasing the group population could influence the demonstration of other relationships, especially in the case of parameters where only the trend towards significance was achieved. On the other hand, the size of the group in the presented study did not differ significantly from other studies on related topics.

Additional study limitations are associated with the absence of a control group and the results of MRI examinations performed with only a 1.5-Tesla machine. The use of 3-Tesla MRI could be associated with greater sensitivity to detect silent microembolic brain lesions in the study group.

## 5. Conclusions

Cerebral microembolism assessed by MRI is often found in patients with paroxysmal AF, and its presence and severity are associated with age and a higher CHA2DS2-Vasc score. Coexistence of PFO and coronary artery disease is an additional factor affecting the severity of the lesions. In a population of relatively young patients with AF, without significant cardiovascular loads, cerebral microembolism is not associated with cognitive impairment. It seems that the PVI-RF procedure does not affect the more frequent occurrence of brain microemboli lesions and cognitive abilities.

## Figures and Tables

**Table 1 jcm-10-00565-t001:** Characteristics of the study group.

Characteristic	Study Group (*n* = 74)
Age, y	58.5 (50–63)
Male gender	45 (60.8%)
AF duration:	
● 0–5 y	33 (44.6%)
● 5–10 y	26 (35.1%)
● >10 y	15 (20.3%)
Height, cm	174.2 ± 10.1
Weight, kg	88.7 ± 15.6
BMI, kg/m^2^	29.2 ± 4.4
BSA, m^2^	2.06 ± 0.22
PLT, ×10^3^	185 (162–215)
APTT, sec.	1.4 (1.3–1.6)
INR	1.9 ± 0.4
EHRA score	3 (2–3)
CHA_2_DS_2_-Vasc score	2 (1–3)
PFO	20 (27%)
Echogenic blood in the left atrium appendage:	
● Absent	63 (85.1%)
● 1 grade	9 (12.2%)
● 2 grade	2 (2.7%)
Coronary artery disease	15 (20.3%)
Arterial hypertension	50 (67.6%)
Diabetes mellitus	15 (20.3%)
Hyperlipidemia	55 (67.6%)
Obesity	26 (35.1%)
Tobacco smoking:	
● Never	44 (59.5%)
● In the past	21 (28.4%)
● Active	9 (12.2%)

Results are given as the mean with standard deviation for normal distributions or the median with interquartile range for non-normal distributions or absolute numbers and percentages. AF—atrial fibrillation, APTT—activated partial thromboplastin time, BMI—body mass index, BSA—body surface area, INR—international normalized ratio, PFO—patent foramen ovale, PLT—blood platelets, y—years.

**Table 2 jcm-10-00565-t002:** WMH brain lesions severity in MRI before PVI-RF.

Characteristic	WMH Absent (*n* = 19)	Fazekas 1 Grade (*n* = 48)	Fazekas 2 Grade (*n* = 4)	Fazekas 3 Grade (*n* = 3)	*p* Value
MMSE	30 (28–30)	29 (28–30)	29 (27–30)	29 (29–30)	0.31
Age, y	44 (38–56)	60 (56.5–63.5)	65,5 (56–69)	60 (40–66)	0.0005
CHA_2_DS_2_-Vasc score	1 (0–2)	2 (1–3)	3 (2–3)	1 (1–4)	0.04
PFO	7 (36.8%)	8 (17%)	2 (50%)	3 (100%)	0.006
CAD	1 (5.3%)	11 (22.9%)	3 (75%)	0 (%)	0.01
Arterial hypertension	10 (52.6%)	34 (70,8)	4 (100%)	2 (66.7%)	0.25
Diabetes mellitus	2 (10.5%)	12 (25%)	0 (0%)	1 (33.3%)	0.37
Hyperlipidemia	11 (57.9%)	34 (70.8)	4 (100%)	1 (33.3%)	0.2
Obesity	4 (21.1%)	19 (39.6%)	1 (25%)	2 (66.7%)	0.31
Tobacco smoking:					
• Never	11 (57.9%)	30 (62.5%)	3 (75%)	0 (0%)	0.1
• In the past	5 (26.3%)	14 (29.2)	2 (25%)	1 (33.3%)	0.1
• Active	3 (15.8%)	4 (8.3)	0 (%)	2 (66.7%)	0.1
AF duration:					
• 0–5 y	8 (42.1%)	25 (52.1%)	0 (%)	0 (0%)	0.1
• 5–10 y	6 (31.6%)	17 (35.4%)	2 (50%)	1 (33.3%)	0.1
• >10 y	5 (26.3%)	6 (12.5%)	2 (50%)	2 (66.7%)	0.1
BMI	27.1 (26.2–29.7)	29.2 (26.3–33)	26.7 (25.9–30.7)	32.5 (25.2–35.1)	0.37

Results are given as the median with interquartile range for non-normal distributions or absolute numbers and percentages. AF—atrial fibrillation, BMI—body mass index, CAD—coronary artery disease, MMSE—Mini-Mental State Examination, MRI—magnetic resonance imaging, PFO—patent foramen ovale, PVI-RF—radiofrequency pulmonary vein isolation, WMH—white matter hiperintensities, y—years.

**Table 3 jcm-10-00565-t003:** WMH brain lesions severity in MRI after PVI-RF.

Characteristic	WMH Absent (*n* = 14)	Fazekas 1 Grade (*n* = 40)	Fazekas 2 Grade (*n* = 6)	Fazekas 3 Grade (*n* = 2)	*p* Value
MMSE	30 (28–30)	29 (28–30)	27 (26–29)	29.5 (29–30)	0.15
Age, y	45 (39–56)	61 (56–64)	65 (58–66)	53 (40–66)	0.001
CHA_2_DS_2_-Vasc score	1 (1–2)	2 (1–3)	3 (3–3)	2.5 (1–4)	0.02
PFO	5 (35.7%)	7 (18%)	2 (33.3%)	2 (100%)	0.05
CAD	1 (7.1%)	9 (22.5%)	4 (66.7%)	0 (0%)	0.03
Arterial hypertension	8 (42.9%)	28 (70%)	6 (100%)	2 (100%)	0.2
Diabetes mellitus	1 (7.1%)	10 (25%)	0 (0%)	1 (50%)	0.19
Hyperlipidemia	10 (71.4%)	26 (65%)	6 (100%)	1 (50%)	0.34
Obesity	2 (14.3%)	15 (37,5%)	1 (16,7%)	1 (50%)	0.31
Tobacco smoking:					
● Never	7 (50%)	27 (67.5%)	4 (66,7%)	0 (0%)	0.36
● In the past	4 (28.6%)	9 (22.5%)	2 (33.3%)	1 (50%)	0.36
● Active	3 (21.4%)	0 (0%)	0 (0%)	1 (50%)	0.36
AF duration:					
● 0–5 y	5 (35.7%)	20 (50%)	1 (16.7%)	0 (0%)	0.3
● 5–10 y	4 (28.6%)	15 (37.5%)	3 (50%)	1 (50%)	0.3
● >10 y	5 (35.7%)	5 (12.5%)	2 (33.3%)	1 (50%)	0.3
BMI	27 (26.2–29.4)	29.2 (26.5–32.7)	26.1 (25.5–27.1)	30.2 (25.2–35.1)	0.2
PVI-RF effective (*n* = 37)	6 (19.4%)	21 (67.7%)	3 (9.7%)	1 (3.2%)	0.9
PVI-RF ineffective (*n* = 35)	8 (26.7%)	18 (60%)	3 (10%)	1 (3.3%)	0.9

Results are given as the median with interquartile range for non-normal distributions or absolute numbers and percentages. AF—atrial fibrillation, BMI—body mass index, CAD—coronary artery disease, MMSE—Mini-Mental State Examination, MRI—magnetic resonance imaging, PFO—patent foramen ovale, PVI-RF—radiofrequency pulmonary vein isolation, WMH—white matter hiperintensities, y—years.

**Table 4 jcm-10-00565-t004:** WMH lesions and psychological assessment before and after PVI-RF.

Characteristic	Before PVI-RF *n* = 74	After PVI-RF *n* = 62	*p* Value
WMH presence	55 (74.3%)	48 (77.4%)	0.1
WMH severity	1 (0–1)	1 (1–1)	0.1
WMH severity:			
● Fazekas 0, grade	19 (25.7%)	14 (22.6%)	
● Fazekas 1, grade	48 (64.9%)	40 (64.5%)	0.79
● Fazekas 2, grade	4 (5.4%)	6 (9.7%)	
● Fazekas 3, grade	3 (4%)	2 (3.2%)	
MMSE, score	29 (27–29.5)	29 (28–30)	0.08

Results are given as the median with interquartile range for non-normal distributions or absolute numbers and percentages. MMSE—Mini-Mental State Examination, MRI—magnetic resonance imaging, PVI-RF—radiofrequency pulmonary vein isolation, WMH—white matter hiperintensities.

## Data Availability

The data presented in this study are available on reasonable request from the corresponding author.

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
