# Peer review of "Hyperintense Brain Lesions in Asymptomatic Low Risk Patients with Paroxysmal Atrial Fibrillation Undergoing Radiofrequency Pulmonary Vein Isolation"

_jcm, 2021, doi:10.3390/jcm10040565_

Round 1

Reviewer 1 Report

  1. "Methods: Eighty patients with AF (58 years (IQR 50-63), 50 male) were included" do you mean "mean 58 years" or median?
  2. Keywords: atrial fibrillation, magnetic resonance imaging, white matter hyperintensities, cognitive 27 decline, Mini-Mental State Examination. Alphabetical order?

  3. "Atrial fibrillation (AF) leads to the formation of small microemboli embolizing cere-31 bral microcirculation." add ref, delete "small" as you already state "small".

  4. Why only 80 patients during 2013-2017? How did you select them? Consecutive patients?
  5. "A p value of 218 less than 0.05 was considered to indicate statistical significance." remove the word "indicate"

  6. Table 1, "mean age"?
  7. Table 3, use "." not "," för p-values, i.e. 0.15 etc.
  8. The absence of a Control Group is a major limitation, which is stated.
  9. Do not have 1,2,3 in Conclusion, write it together.

Author Response

Review 1

"Methods: Eighty patients with AF (58 years (IQR 50-63), 50 male) were included" do you mean "mean 58 years" or median?

I mean median age. The age had non-normal distribution, therefore it has been presented as the median with IQR. I am aware that this is rare. I have made correction to the abstract. Line 15.

Keywords: atrial fibrillation, magnetic resonance imaging, white matter hyperintensities, cognitive 27 decline, Mini-Mental State Examination. Alphabetical order?

I have made corrections to the manuscript. Lines 27-29.

"Atrial fibrillation (AF) leads to the formation of small microemboli embolizing cerebral microcirculation." add ref, delete "small" as you already state "small".

I have made corrections to the manuscript. Lines 31-32.

Why only 80 patients during 2013-2017? How did you select them? Consecutive patients?

Patients were enrolled from 2013 to the end of 2015, the follow up lasted until 2017. The small number of enrolled patients is due to the restrictive inclusion criteria. The population consists of relatively young patients with a history of paroxysmal, symptomatic AF (median EHRA 3), without significant structural heart disease and with a low score obtained in the CHA2DS2-Vasc scale that were classified to radiofrequency pulmonary vein isolation. Patients who had undergone any pulmonary vein isolation, brain ischemic events or had contraindications for brain MRI were disqualified.

"A p value of less than 0.05 was considered to indicate statistical significance." remove the word "indicate"

I have made corrections to the manuscript. Lines 218-219.

Table 1, "mean age"

As mensioned above, age had non-normal distribution, therefore it has been presented as the median interquartile range. Table 1, line 235.

Table 3, use "." not "," för p-values, i.e. 0.15 etc.

I have made corrections to the manuscript. Table 3, line 299.

The absence of a Control Group is a major limitation, which is stated.

I fully agree that referring to a control group would give a broader view of the topic. At the moment, however, we plan to expand the study group first. This would require an additional bioethics committee approval, while from experience I will say that patients are reluctant to undergo brain MRI, especially twice a year as in the presented study.

Do not have 1,2,3 in Conclusion, write it together.

I have made corrections to the manuscript. Lines 495-502.

Reviewer 2 Report

The paper is of significant clinical interest and adds elements of interest to the literature on the subject. The article is well presented. However, some methodological and statistical aspects need to be better specified and improved. 1- the percentage of WMH lesions after PVI is not reported in the abstract. This is one of the most interesting results of the paper. 2- Crucial Point !: The authors state that this is a prospective study in which all 80 patients underwent MRI. (line 142). However, in tables 2 and 4 the total number of patients undergoing MRI is 74. Furthermore, the authors state that 55 patients (68.8% out of 80 patients?) Showed the presence of WMH before PVI (actually 55 out of 74 patients are 74.3 %). The post PVI result is correctly calculated for 62 patients who actually underwent MRI. The authors should explain why some patients did not underwent MRI before ablation and were equally included in the series and correct the results. 3- The authors show the various predictors of WMH lesions before and after PVI ablation in univariate analisys. Multivariate analysis is strongly recommended. 4- include in table 2 (pre pvi) and 3 (post PVI), BMI which is predictive only post pvi, moreover other analyzed variables should be included . 5- Also include in table 3 the groups with effective and ineffective PVI and the incidence of WMH. 6- the discussion can be shortened by focusing on the focal points of the results (see conclusions) 7- minor point: the Thermo cool smart Touch SF catheter was only available from 2016. Previously the non-SF model was available.

Author Response

1- the percentage of WMH lesions after PVI is not reported in the abstract. This is one of the most interesting results of the paper.

This information was not included in the abstract due to the restrictive word limit. I have made correction to the abstract. Line 18.

2- Crucial Point !: The authors state that this is a prospective study in which all 80 patients underwent MRI. (line 142). However, in tables 2 and 4 the total number of patients undergoing MRI is 74. Furthermore, the authors state that 55 patients (68.8% out of 80 patients?) Showed the presence of WMH before PVI (actually 55 out of 74 patients are 74.3 %). The post PVI result is correctly calculated for 62 patients who actually underwent MRI. The authors should explain why some patients did not underwent MRI before ablation and were equally included in the series and correct the results.

I fully agree with the remark. Thank you for noticing my mistake. The paper is a part of series of analyses based on the same patients group. This time only patients who underwent brain MRI should be included. Whole group characteristics is corrected and recalculated for the patients who underwent MRI. All other presented results include only patients with known MRI result. Statistical comparisons were done with exclusion of data gaps.

I have made corrections to the table (table 1, line 235) and manuscript (lines 15, 105-106, 230, 243-249).

3- The authors show the various predictors of WMH lesions before and after PVI ablation in univariate analisys. Multivariate analysis is strongly recommended.

We did calculation of logistic regression for dichotomic variable – WMH present or not (WMH occurrence). In the whole group, before PVI-RF procedure and after follow-up period multivariate logistic regression was performed with the inclusion of the following variables: CHA2DS2-Vasc, AF duration time, BMI, PFO presence, echogenic blood presence in the left atrium appendage, tobacco smoking and hyperlipidemia. In both cases only CHA2DS-Vasc was statistically significant (p=0.02). In the second one multivariate logistic regression analysis model we took into account the variables included in the CHA2DS2-Vasc scale: arterial hypertension, age, diabetes mellitus, sex, coronary artery disease and peripheral artery disease (patients with congestive heart failure were excluded from the study). In this case only age was statistically significant for both: before PVI-RF procedure and after follow-up period (p<0.001).

The above results were included in the manuscript. Lines 274-290 and 311-318.

4- include in table 2 (pre pvi) and 3 (post PVI), BMI which is predictive only post pvi, moreover other analyzed variables should be included.

I have made corrections to the tables. However, I would like to emphasize that BMI had a statistically significant influence only on the presence of brain WMH lesions, with no impact on their severity. Tables 2 (line 267) and table 3 (line 299).

5- Also include in table 3 the groups with effective and ineffective PVI and the incidence of WMH.

I have made corrections to the table. Table 3, line 299.

6- the discussion can be shortened by focusing on the focal points of the results (see conclusions)

As suggested, the discussion has been shortened by removing duplicate statements or unrelated to the leading topic. Lines 342 – 478.

7- minor point: the Thermo cool smart Touch SF catheter was only available from 2016. Previously the non-SF model was available.

I have made corrections to the manuscript. Line 201.

Round 2

Reviewer 1 Report

No more comments 

Reviewer 2 Report

The authors made the requested changes correctly.